# Early ICD implantation following out-of-hospital cardiac arrest: a retrospective cohort study from the Swedish Registry for Cardiopulmonary Resuscitation

Pedram Sultanian ,[1] Peter Lundgren,[1,2] Aidin Rawshani,[1] Sebastian Möller,[2] Arash Hadi Jafari,[3] Laura David,[2] Shavan Yassinson,[4] Anna Myredal,[1] Cecilia Rorsman,[5] Amar Taha,[2] Annica Ravn-Fischer,[6] Andreas Martinsson,[2] Johan Herlitz,[7] Araz Rawshani [1]

For numbered affiliations see end of article.

**Correspondence to**
Pedram Sultanian;
pedram_sultanian@hotmail.com

## ABSTRACT

**Background** It is unclear whether an implantable cardioverter-defibrillator (ICD) is generally beneficial in survivors of out-of-hospital cardiac arrest (OHCA).
**Objective** We studied the association between ICD implantation prior to discharge and survival in patients with cardiac aetiology or initial shockable rhythm in OHCA.
**Design** We conducted a retrospective cohort study in the Swedish Registry for Cardiopulmonary Resuscitation. Treatment associations were estimated using propensity scores. We used gradient boosting, Bayesian additive regression trees, neural networks, extreme gradient boosting and logistic regression to generate multiple propensity scores. We selected the model yielding maximum covariate balance to obtain weights, which were used in a Cox regression to calculate HRs for death or recurrent cardiac arrest.
**Participants** All cases discharged alive during 2010 to 2020 with a cardiac aetiology or initial shockable rhythm were included. A total of 959 individuals were discharged with an ICD, and 2046 were discharged without one.
**Results** Among those experiencing events, 25% did so within 90 days in the ICD group, compared with 52% in the other group. All HRs favoured ICD implantation. The overall HR (95% CI) for ICD versus no ICD was 0.38 (0.26 to 0.56). The HR was 0.42 (0.28 to 0.63) in cases with initial shockable rhythm; 0.18 (0.06 to 0.58) in non-shockable rhythm; 0.32 (0.20 to 0.53) in cases with a history of coronary artery disease; 0.36 (0.22 to 0.61) in heart failure and 0.30 (0.13 to 0.69) in those with diabetes. Similar associations were noted in all subgroups.
**Conclusion** Among survivors of OHCA, those discharged with an ICD had approximately 60% lower risk of death or recurrent cardiac arrest. A randomised trial is warranted to study this further.

## STRENGTHS AND LIMITATIONS OF THIS STUDY

⇒ Large and comprehensive data set: the study included all cases enrolled in a nationwide registry, with very high coverage, including cases over a long period (2010–2020).
⇒ State-of-the-art data analysis: we used a comprehensive data-driven machine learning approach to generate propensity scores, and based our estimates on the score yielding maximum covariate balance.
⇒ Retrospective cohort study, the nature of data: this study is a retrospective cohort study which limits causal inferences.
⇒ Timing of implantable cardioverter-defibrillator (ICD) implantation: ICD at discharge was defined as having an ICD at hospital discharge, without specifying when the ICD had been implanted, such that the results should be interpreted as the association between having an ICD at discharge and survival.

## INTRODUCTION

An implantable cardioverter-defibrillator (ICD) is indicated in primary and secondary prevention of sudden cardiac arrest (SCA) from ventricular fibrillation (VF) or ventricular tachycardia (VT). ICDs are considered in patients with persisting significant risk of VT/VF if guideline-directed medical therapy (GDMT) and invasive procedures are insufficient to eliminate future risk of VF/VT. Several randomised trials have compared ICD with GDMT for primary prevention of SCA. With regard to secondary prevention, the antiarrhythmics versus implantable defibrillators (AVID) demonstrated that an ICD is superior to amiodarone for preventing a recurrent cardiac arrest.[1 2] Guidelines currently recommend that an ICD should be considered in cases without transient or reversible causes of SCA due to VF/VT, haemodynamically unstable VT, sustained VT, provided that life-expectancy is 1 year or longer.

It is still unclear whether SCA and ventricular arrhythmias during an acute myocardial infarction should prompt ICD implantation. It is currently believed that such cases are not

at significant future risk of SCA. However, the underlying substrate (myocardial scar tissue) may persist even if revascularisation is successful. A range of factors that are not routinely assessed may influence future risk of ventricular arrhythmias.[1–8] This was presumably demonstrated in the AVID registry, which showed that patients not randomised to ICD, with transient or reversible causes of SCA, had comparable or possibly inferior outcomes compared with those randomised to ICD.[7]

The aim of this study was to study the association between having an ICD at discharge and survival among patients surviving an out-of-hospital cardiac arrest (OHCA) due to cardiac aetiology or with an initial shockable rhythm.

## METHODS

All cases enrolled in the Swedish Registry for Cardiopulmonary Resuscitation (SRCR) from 1 January 2010 to 11 June 2020 were assessed (these dates were arbitrarily chosen to include roughly 10 years, and we did not have data beyond the dates the ethical approval was submitted). This included a total of 54 956 cases of OHCA, of whom 54 568 were first events. We excluded all recurring events (n=388), such that only the first OHCA was assessed for each individual. A total of 10 836 cases had information on ICD, of whom 3851 were discharged alive. Among these 3851 patients, we included those with cardiac aetiology or initial shockable rhythm, resulting in a final study population of 3005 patients.

### Exposure

Implantation of an ICD during hospitalisation was recorded as a binary variable in the registry.

### Outcomes

The primary outcomes were the composite of death, recurrent OHCA or IHCA (in-hospital cardiac arrest). Mortality data were retrieved from the Swedish Cause of Death Registry, which has complete ascertainment. Recurrent OHCA and new IHCA events were retrieved using the SRCR, with follow-up ending in 11 June 2020.

### Variable selection

A total of 919 candidate predictors were assessed. These included 41 predictors relating to patient phenotype, initial presentation, prehospital and in-hospital management (43 predictors), socioeconomic variables (43 predictors), medications (19 agents), previous coexisting conditions and diagnoses received during index hospitalisation (814 diagnoses). Refer to table 1 and online supplemental table 1 for further details. We removed variables with zero variance, as well as factor variables with >30 levels. With regard to the latter, all variables represented socioeconomic data, which was still adequately represented by remaining variables. The final data set contained 438 candidate predictors of ICD implantation.

### Feature selection

We used both random forest (RF) and gradient boosting (GBM) to identify the 50 most important predictors of

ICD implantation, refer to online supplemental figures 1 and 2. This was done by computing two separate models with all 438 predictors, using ICD implantation as the dependent variable. The random number generator was not fixed in order to ensure reproducibility across computations, which we confirmed. We checked the consistency between RF and GBM, knowing that these methods yield a different variable importance. Thus, we used RF and GBM to identify—according to each algorithm—the top 50 predictors of ICD implantation. These 50 predictors were then used in several models (outlined below) to obtain several propensity scores, in order to find the model, maximising covariate balance.

### Propensity score and inverse probability of treatment weighting (IPTW)

We estimated propensity scores and generated balancing weights to estimate the average treatment effect (ATE) and average treatment effect on the treated (ATT). ATE measures the average treatment effect in the entire population, should the entire population receive the treatment. ATT measures the treatment effect within the group of patients who actually received the treatment. Interested readers are referred to Greifer and Stuart *et al*.[8] Weights were generated using GBM, extreme gradient boosting (XGBOOST), Bayesian additive regression trees (BART), neural networks and logistic regression. Each modelling framework was used to model ICD implantation using the 50 top predictors identified by GBM as well as RF, as explained above. Tree-based methods were tuned with regard to tree depth, number of trees and learning rate. Tuning was optimised to minimise covariate standardised mean difference (SMD). SMD below 0.1 were considered non-consequential.[9] The weights were used in Cox proportional hazards models to calculate HRs for the endpoints (death, recurrent OHCA and IHCA) from discharge to end of follow-up. Subgroup analyses were performed in relation to age, sex, comorbidities and patient characteristics relating to the event.

The purpose of our approach to estimate the final propensity score was to consider all available variables, identify the most important ones and then consider several different prediction models to finally obtain the maximum covariate balance. Ultimately, this allows us to estimate the effect of having an ICD at discharge, on the risk of the outcomes (composite of death, recurrent OHCA or IHCA), as can be estimated using observational data.

### Patient and public involvement

None.

### Ethical consideration

The study was approved by the Swedish Ethical Review Authority (#2020–02017). The Ethical Review Authority exempted the need for informed consent, due to the retrospective nature of data.

**Table 1** Baseline characteristics (abbreviated) in patients who were discharged after out-of-hospital cardiac arrest, in relation to ICD implantation

| Variable | No ICD 2046 | ICD 959 | P value | SMD |
|---|---|---|---|---|
| Women, n (%) | 435 (21.3) | 173 (18.1) | 0.046 | 0.081 |
| Age, mean (SD) | 63.6 (13.7) | 58.5 (15.4) | <0.001 | 0.347 |
| **Coexisting conditions prior to OHCA** | | | | |
| Hypertension | 607 (29.7) | 292 (30.4) | 0.694 | 0.017 |
| Ischaemic heart disease | 280 (13.7) | 242 (25.2) | <0.001 | 0.295 |
| Stable or unstable angina | 213 (10.4) | 168 (17.5) | <0.001 | 0.206 |
| Heart failure | 181 (8.8) | 174 (18.1) | <0.001 | 0.275 |
| Acute myocardial infarction | 178 (8.7) | 161 (16.8) | <0.001 | 0.244 |
| Type 2 diabetes | 164 (8.0) | 114 (11.9) | 0.001 | 0.13 |
| Stroke | 79 (3.9) | 33 (3.4) | 0.643 | 0.022 |
| Renal failure | 73 (3.6) | 32 (3.3) | 0.83 | 0.013 |
| **Medications prior to OHCA** | | | | |
| RAAS blockers | 548 (26.8) | 339 (35.3) | <0.001 | 0.186 |
| Anticoagulants | 405 (19.8) | 331 (34.5) | <0.001 | 0.336 |
| Beta-blockers | 434 (21.2) | 293 (30.6) | <0.001 | 0.214 |
| Lipid lowering drugs | 349 (17.1) | 267 (27.8) | <0.001 | 0.261 |
| **Location of cardiac arrest** | | | <0.001 | 0.477 |
| Home | 832 (40.7) | 411 (42.9) | | |
| Public place | 644 (31.5) | 447 (46.6) | | |
| Other places | 568 (27.8) | 101 (10.5) | | |
| **Prehospital interventions** | | | | |
| Mechanical compressions | 514 (26.2) | 257 (27.7) | 0.416 | 0.034 |
| Intubation | 295 (14.7) | 147 (15.6) | 0.591 | 0.023 |
| Defibrillated, any | 1738 (85.8) | 867 (91.1) | <0.001 | 0.165 |
| Epinephrine | 719 (36.0) | 411 (43.8) | <0.001 | 0.161 |
| Amiodarone | 352 (17.7) | 246 (26.4) | <0.001 | 0.211 |
| **Critical time intervals, min—median (IQR)** | | | | |
| Arrest to CPR start | 0.00 (0.00, 2.00) | 1.00 (0.00, 4.00) | <0.001 | 0.18 |
| Arrest to first defibrillation | 6.00 (1.00, 12.00) | 10.00 (6.00, 13.00) | <0.001 | 0.33 |
| Arrest to ambulance arrival | 9.00 (6.00, 13.00) | 9.00 (7.00, 14.00) | 0.021 | 0.095 |
| Arrest to ROSC | 11.00 (4.00, 18.00) | 13.00 (8.50, 21.00) | <0.001 | 0.263 |
| **Initial rhythm** | | | 0.001 | 0.167 |
| VF/pVT | 1667 (92.6) | 852 (96.2) | | |
| PEA | 65 (3.6) | 12 (1.4) | | |
| Asystole | 69 (3.8) | 22 (2.5) | | |
| Circulation on hospital arrival | 1932 (95.3) | 913 (95.8) | 0.614 | 0.024 |
| Conscious on hospital arrival | 1021 (50.9) | 345 (36.8) | <0.001 | 0.288 |
| **Circumstances at time of arrest** | | | | |
| Witnessed arrest | 1879 (92.7) | 866 (91.4) | 0.283 | 0.045 |
| Bystander CPR | 1080 (54.4) | 743 (78.6) | <0.001 | 0.531 |
| **Inhospital interventions** | | | | |
| PCI | 1495 (74.0) | 364 (38.9) | <0.001 | 0.756 |
| CABG | 99 (4.9) | 34 (3.6) | 0.148 | 0.063 |

AED, automated external defibrillator; CABG, coronary artery bypass grafting; CPR, Cardiopulmonary Resuscitation; ECMO, extracorporeal membrane oxygenation; EMS, emergency medical system; EU, European Union; ICD, implantable cardioverter defibrillator; OHCA, out-of-hospital cardiac arrest; PCI, percutaneous coronary intervention; PEA, pulseless electrical activity; PVT, pulseless ventricular tachycardia; RAAS, Renin-Angiotensin-Aldosterone System; ROSC, return of spontaneous circulation; SMD, standardised mean difference, is calculated as the difference in means between the two groups, divided by their combined SD. Balance was defined as SMD<0.1; VF, ventricular fibrillation.

## RESULTS

### Baseline characteristics

A total of 3005 survivors of OHCA were included, of whom 959 had an ICD implanted and 2046 were discharged without an ICD. Mean age was 58.5 years and 63.6 years in cases with and without ICD implantation, respectively. Women constituted 21% of cases without ICD implanted, compared with 18% of those with an ICD implanted (table 1).

A history of hypertension was equally prevalent. Previous ischaemic heart disease, dyslipidaemia, angina pectoris, atrial fibrillation, heart failure, myocardial infarction and diabetes were more prevalent prior to OHCA in cases who received an ICD. Cardiovascular medications were also more common in patients who received an ICD (table 1).

Pre-hospital defibrillations were performed in 91.1% of cases who received an ICD, compared with 85.8% of those who did not receive an ICD. Epinephrine and amiodarone were more frequently administered in cases who received an ICD. The initial rhythm was shockable in 96.2% of cases who received an ICD, compared with 92.6% of cases who did not. The vast majority (95.8% and 95.3% in those with and without ICD, respectively) in both groups displayed signs of spontaneous circulation at arrival to hospital. PCI was performed more frequently in those who did not receive an ICD (table 1).

### Survival

A total of 320 deaths, 138 recurrent OHCAs and 70 IHCAs occurred after discharge. The most pronounced survival trend was noted for cases who did not receive an ICD, of whom the majority of deaths occurred early after discharge (figure 1). In men and women, 45% and 58%, respectively, of those who experienced the composite endpoint did so within 90 days, with the majority doing so within 30 days. Among those experiencing events, 25% did so within 90 days in the ICD group, compared with 52% within 90 days among the others.

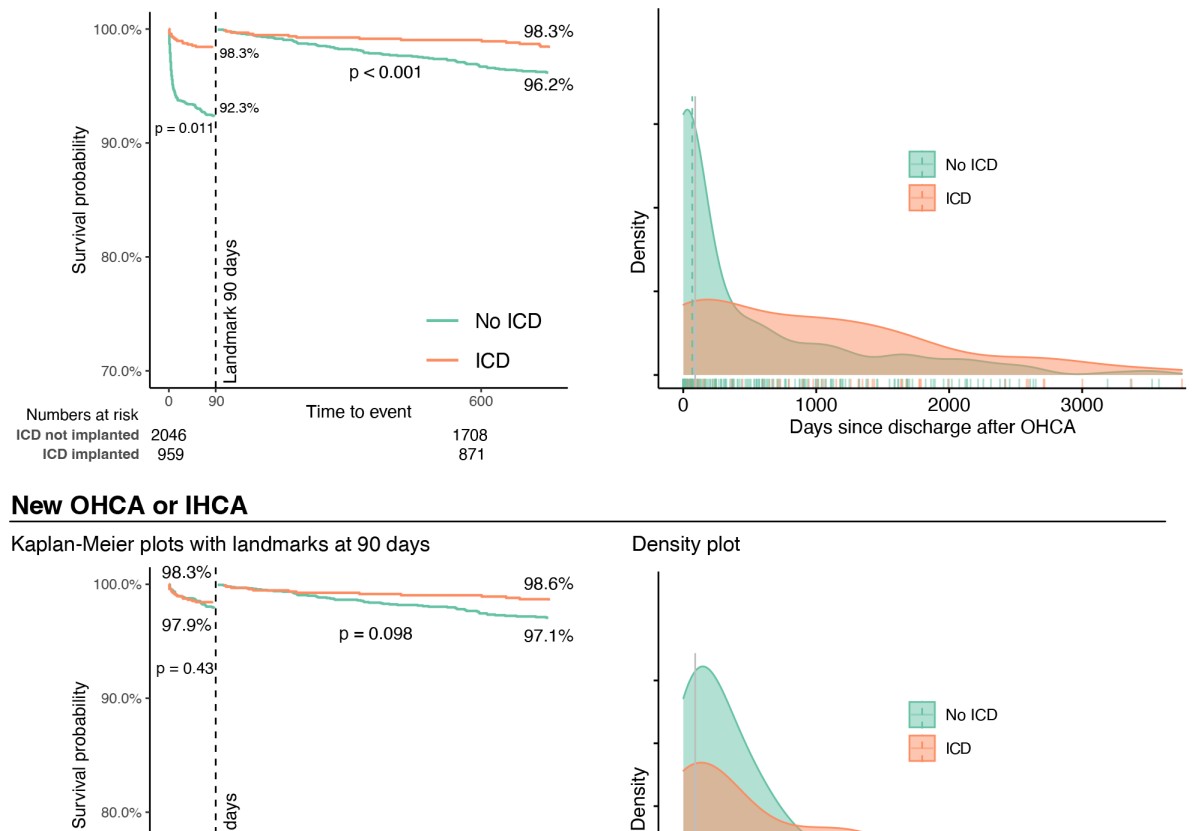

**Figure 1**  Kaplan-Meier plots (left) with 90-day cut-off landmark, and density plots (right) for the combined endpoints. Survival distributions after the landmark include only those surviving upto 90 days.

## Modelling results

RF and GBM identified (in separate models) 50 predictors with greatest influence in ICD implantation. The predictors not shared by both frameworks were the following: educational level, individual income, time from emergency dispatch to EMS arrival, clock at time of cardiac arrest, use of mechanical compressions, intubation, use of laryngeal intubation, marital status, alcohol misuse,

type 2 diabetes, children aged 4–6 and 11–15 at home and use of calcium channel blockers.

## Covariate balance

Covariate balance obtained using GBM and top 50 predictors identified by RF yielded the best covariate balance, as judged by differences in standardised mean difference, coefficient of variation and distribution of weights. BART

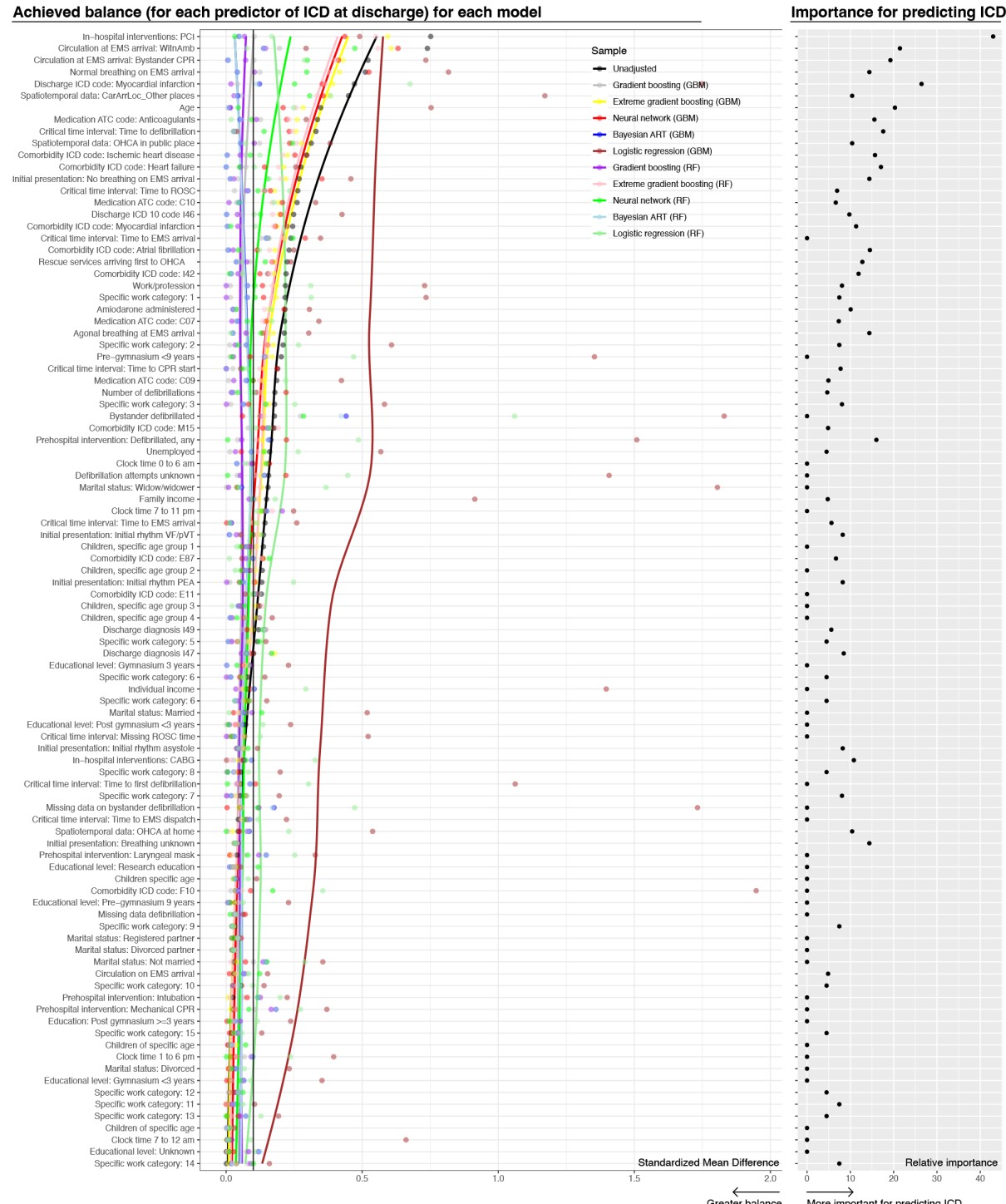

**Figure 2** Covariate balance obtained by 10 different modeling strategies. Left panel: covariate balance as assessed by standardised mean differences. Right panel: Relative importance of each variable, in terms of predicting ICD at discharge.

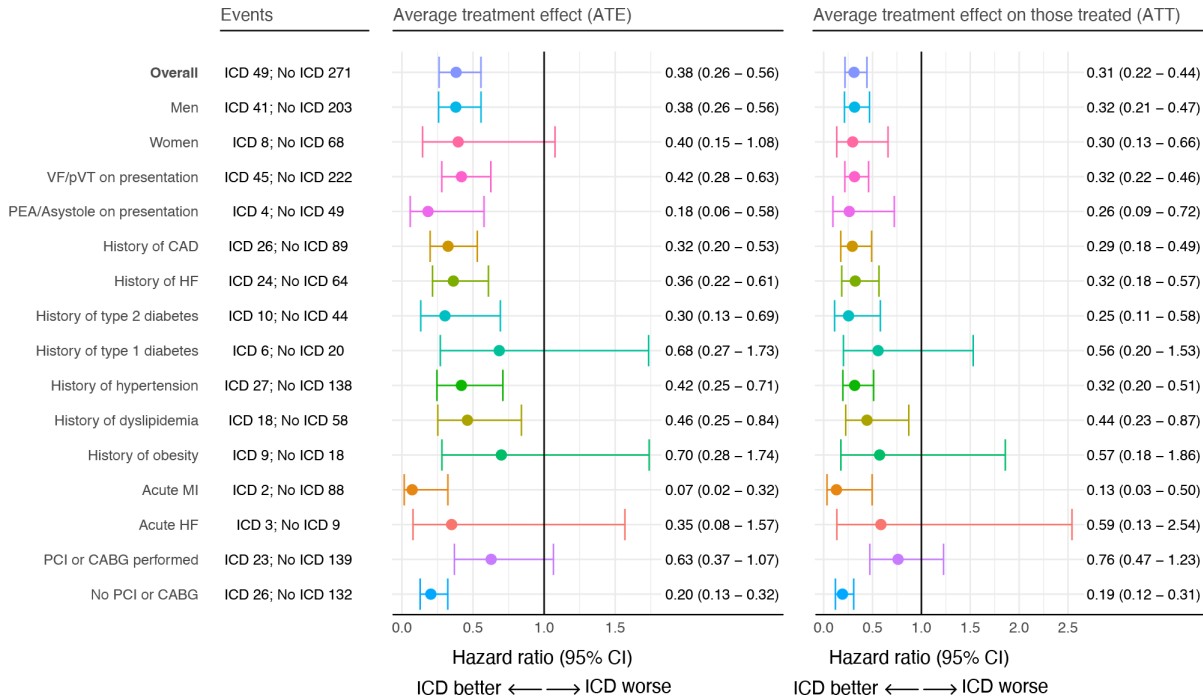

**Figure 3** HRs for death, recurrent OHCA or IHCA, using the final model. IHCA, in-hospital cardiac arrest; OHCA, out-of-hospital cardiac arrest.

yielded the second best balance. Figure 2 displays the unadjusted and weight-adjusted covariate balance across all 10 models (left panel), along with the RF-derived variable importance (right panel). After adjusting for weights, the GBM model had not balanced the following covariates: history of arthrosis, breathing at EMS arrival, OHCA witnessed by ambulance, use of anticoagulants prior to OHCA, place of cardiac arrest and arrhythmias during hospitalisation. The latter was the 41st most important predictor, and included diagnoses such as sick sinus syndrome, brady–tachy arrhythmias, premature beats and VF (ICD-10 code I.49). That variable was not among the 50 most important predictors for the GBM model.

### Association between ICD implantation and outcomes
All point estimates, for both ATE and ATT analyses were below 1.0, with the vast majority being significant, favouring ICD implantation (figure 3).

With regard to ATE, the overall HR (95% CI) for ICD versus no ICD was 0.38 (0.26–0.56). The point estimate was 0.40 among women, although not statistically significant (eight events occurred in females with ICD). For cases with VF/pVT (pulseless ventricular tachycardia) as initial rhythm, HR was 0.42 (0.28–0.63). For cases with PEA/asystole, HR was 0.18 (0.06–0.58). HR for those with a history of CAD, heart failure and diabetes were 0.32 (0.20–0.53), 0.36 (0.22–0.61), 0.30 (0.13–0.69), respectively. The lowest point estimates were noted for cases with an acute MI, PEA/asystole on presentation and those who did not undergo PCI or CABG.

With regard to ATT, the HRs for ICD versus no ICD were aligned with those obtained for ATE.

With regards to non-balanced covariate, adding them to the Cox regression did not affect any point estimate (online supplemental figure 3).

### DISCUSSION
Our results clearly demonstrate that those discharged with an ICD were at substantially lower risk of death or recurrent cardiac arrest, with all HRs around 0.3, suggesting a 70% reduction in the probability of death or recurrent arrest. While these results encourage a broader use of ICDs, one cannot exclude confounding by indication given the observational nature of the data. We found that the highest incidence rate for death or recurrent cardiac arrest was seen early after discharge (figure 1). Aligned with these results, a Danish study (concluded during 2007–2011) found that patients with early ICD implantation (during hospitalisation) had a higher survival probability postdischarge, with HRs similar to ours; HR for death in the ICD group versus no ICD was 0.44 (95% CI: 0.23 to 0.88).[10] Other studies have presented similar trends in results.[1 2 10 11]

Current clinical guidelines suggests that patients suffering from ventricular arrythmias due to a resolving cardiac event do not have an increased risk for future SCA. Thus, excluding these patients from receiving treatment with an ICD. However, previous reports have suggested otherwise, showing these individuals have an elevated risk for future malignant arrhythmias.[2 7 8] These ideas are supported by and described in the AVID study.[1] In this study, we approached this issue by examining

the ATE and the ATT in cases with OHCA of presumed cardiac aetiology, or with an initial shockable rhythm.

European clinical guidelines recommend ICD implantation for patients with malignant arrhythmias after an acute myocardial infarction with pre-existing left ventricular dysfunction or incomplete revascularisation (*class IIb C indication*).[12–16] ICD implantation is recommended regardless of aetiology for secondary prevention in patients with dilated cardiomyopathy or hypokinetic non-dilated cardiomyopathy in survivors of OHCA with an initial shockable rhythm, or with haemodynamically significant monomorphic VT (*class I recommendation*).[16] After a cardiac arrest, patients are followed at an outpatient clinic and if left ventricular ejection fraction is not restored adequately, then an ICD is recommended. However, our study suggests that this initial time period after discharge carries the highest risk for these patients and future guidelines should take this into consideration.[11]

In this study, we also performed 15 subgroup analyses (figure 3) for both ATE and ATT. All analyses demonstrated similar results, with a strong reduction in the probability of death or recurrent cardiac arrest in individuals receiving an ICD. Notable exceptions to this was the non-significance in the subgroups consisting of women, type 1 diabetes, obesity, acute heart failure and those undergoing PCI or CABG. With regard to women, although the ATE estimate was not significant, the ATT was clearly significant (HR: 0.30 (95% CI: 0.13 to 0.66)). Regarding individuals with type 1 diabetes, obesity, acute heart failure, PCI or CABG performed, it is likely that we were underpowered to assess these subgroups.

Another interesting finding was that PEA/asystole on presentation was associated with a lower risk of outcome in the ICD group. This may be explained by the fact that ambulance response times are increasing which is strongly associated with reductions in the rate of cases presenting with shockable rhythms.[17] Hence, a significant proportion of those with asystole may have VF/pVT as their instigating event, but deteriorate to non-shockable rhythms by the time the first ECG is recorded. Moreover, in these cases, we do not have data on subsequent (after ROSC) occurrences of ventricular arrhythmias, which would increase their likelihood of receiving an ICD.

This study is a registry-based study, which hampers our ability to draw causal inferences. We do not have access to exact timing of ICD implantation, and it is possible that some individuals may have had ICDs implanted prior to cardiac arrest. However our intention-to-treat design emphasises on having an ICD at discharge. We do not have information on in-hospital ECG monitoring data. Residual confounding and confounding by indication are important caveats of propensity score methods. Confounding by indication in this context implies that the propensity score method may be unable to completely remove confounders related to the decision to implant an ICD (eg, patients having longer expected survival, etc). Even with these considerations, the propensity score, especially when generated through machine

learning techniques, stands out as a leading method for imitating a randomised controlled trial using observational data.[9 18] This approach is especially pertinent when utilising real-world data that encompass the whole population targeted for inferences, as in this national registry. With regard to missing data, we found that the rate for ICD implantation was around 15% among survivors with cardiac aetiology. Missingness may affect the robustness of our estimates, although we argue that the pronounced and consistent reduction of the outcomes cannot be explained by missingness which is more likely to be due to randomness.

The DanICD study (NCT ID NCT04576130) will contribute valuable evidence to this topic. It is a randomised study with the aim to assess whether there is a benefit of ICD-implantation in patients with coronary artery disease (including acute myocardial infarction), who survive cardiac arrest due to VF/sustained VT and undergo revascularisation and with an LVEF greater than 35%.

## CONCLUSION

Our study suggests that, on a population and individual level, broad use of ICDs are associated with markedly lower risk of death and recurrent cardiac arrest after OHCA. ICD therapy was associated with approximately 60% reduction in the probability of these outcomes. Randomised trials are warranted to further unravel the optimal use of ICDs post cardiac arrest.

**Author affiliations**
[1]Department of Molecular and Clinical Medicine, Institute of Medicine, Gothenburg, Sweden
[2]Department of Cardiology, Sahlgrenska University Hospital, Goteborg, Sweden
[3]Department of Cardiology, Halmstad Hospital, Halmstad, Sweden
[4]Department of Cardiology, Kungälvs sjukhus, Kungalv, Sweden
[5]Department of Cardiology, Varberg Sjukhus, Varberg, Sweden
[6]Sahlgrenska University Hospital, Institution of Molecular and Clinical Medicine, Gothenburg, Sweden
[7]University of Borås, Faculty of Caring Science, Work Life and Social Welfare, Borås, Sweden

**Contributors** PS and ArR are responsible for the overall content as guarantors of this work. Study concept, design and calculations were done by PS and ArR. First draft of the manuscript was written by PS. ArR ensured data acquisition. PL, AiR, SM, JH, LD, AMy, CR, AT, AR-F, AHJ, SY and AMa contributed significantly to interpretation of data and revision of the manuscript. All authors unanimously agreed on all aspects of the work, including the accuracy and integrity of the manuscript as well as the results. All authors were also involved in responding to reviewer concerns.

**Funding** This work was supported by a generous donation from the Knut and Alice Wallenberg foundation (Rawshani), the Swedish Research Council (Rawshani), Swedish state under the agreement between the Swedish government and the county councils (Rawshani), Gothenburg University (Rawshani) and Region Västra Götaland (Rawshani), to the Wallenberg Center for Molecular and Translational Medicine (WCMTM) in Gothenburg (Rawshani).

**Competing interests** None declared.

**Patient and public involvement** Patients and/or the public were not involved in the design, or conduct, or reporting or dissemination plans of this research.

**Patient consent for publication** Not required.

**Ethics approval** This study involves human participants. Ethics approval for the study has been obtained by the Swedish ethical review authority (#2020-02017). Participants gave informed consent to participate in the study before taking part.

**Provenance and peer review** Not commissioned; externally peer reviewed.

**Data availability statement** Data are available upon reasonable request. Please contact ArR (araz.rawshani@gu.se) for information regarding data availability, which is governed by Swedish law.

**ORCID iDs**
Pedram Sultanian http://orcid.org/0000-0002-6941-6659
Araz Rawshani http://orcid.org/0000-0003-2066-3533

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
