## [Reviewer comments · BMJ Open]

ARTICLE DETAILS

TITLE (PROVISIONAL)	Early ICD implantation following out-of-hospital cardiac arrest – A retrospective cohort study from the Swedish Registry for Cardiopulmonary Resuscitation
AUTHORS	Sultanian, Pedram; Lundgren, Peter; Rawshani, Aidin; Möller, Sebastian; Jafari, Arash Hadi; David, Laura; Yassinson, Shavan; Myredal, Anna; Rorsman, Cecilia; Taha, Amar; Ravn-Fischer, Annica; Martinsson, Andreas; Herlitz, Johan; Rawshani, Araz

VERSION 1 – REVIEW

REVIEWER	Søholm, Helle Copenhagen University Hospital Rigshospitalet, Department of Cardiology
REVIEW RETURNED	21-Aug-2023

GENERAL COMMENTS	Review " Early ICD implantation and survival after out-of-hospital cardiac arrest" Thank you for the opportunity to review the submitted manuscript. The study examines approx. 55.000 patients registered in the SwedHeart registry with OHCA of cardiac cause from 2010 to 2020 and the association between ICD implantation and survival. Approx. 3000 patients with available ICD-data and shockable rhythm were included the final study cohort (patients discharged alive). The result of the study is clear with a HR of 0.38 for ICD-implantation, however taking the observational and retrospective study design into account. General comments: Please describe in further detail how the final study population is defined. It is mentioned that "available ICD-data", however please include data on patients with cardiac cause and shockable rhythm with missing ICD-data? My concern is that a large confounder is introduced. Tables are large with many data presented. Please consider whether some data could be moved to a supplementary table. In the result section more wording is used for mentioning factors not fitted into the models than actually presenting important covariates associated with outcome after ICD-implantation. I suggest rephrasing the results section with more focus on the clinical important factors with less statistical description.
---

	The discussion is section is well-written. Please also include the DanICD study currently enrolling patients (ClinicalTrials.gov ID NCT04576130). Minor comments: The abstract is well-written and nicely presents the scope and results of the manuscript. However, the first sentence of the conclusion “Mortality in survivors of OHCA occurs early after discharge” is not substantiated in the result section. The 90-day event rate is presented, however not how early after discharge death occurs. Please include this in the abstract. A conclusion section in the main manuscript seems to be missing.
--	---

REVIEWER	Reinier, Kyndaron Cedars-Sinai Medical Center, Smidt Heart Institute
REVIEW RETURNED	29-Aug-2023

GENERAL COMMENTS	Summary: This paper uses data from the Swedish Registry for Cardiopulmonary Resuscitation (SRCR) (2010 to 2020) for a total of 3005 patients who survived an out of hospital cardiac arrest (OHCA) with a cardiac etiology or initial shockable rhythm. The authors report that early implantation of ICDs is effective in preventing the combined outcome of overall mortality, out-of-hospital cardiac arrest, or in-hospital cardiac arrest. This is an important topic, and this analysis provides some intriguing data which is potentially useful for guiding clinical practice. However, the manuscript in its current form is somewhat disorganized. Its main methods need more detail, and some results are difficult to follow. The Methods and Results include analysis of features associated with implantation (or non-implantation) of an ICD, which was not described as a main aim of the study. This is, however, an important aspect of this analysis and needs to be incorporated into the overall paper more effectively. Also, machine learning (ML) techniques are used in the analysis that may be less familiar to some readers; better descriptions of these approaches are needed (e.g., see comments re: Figure 2 below). Major comments:  1. The paper’s main aim, as stated in the Abstract and the last paragraph of the Introduction, was to study the impact of early ICD implantation on survival. The sections on predictors of ICD implantation are therefore somewhat unexpected and don’t seem well-integrated into the paper. The authors should consider adding information in the introduction regarding why you are evaluating predictors of ICD implantation and be more explicit about that separate step in the Methods and Results. 2. The “Covariate balance” section in Results could potentially be moved to a supplementary file. Its results do not seem directly related to the study’s main aims. However, a paragraph could be added here that identifies the main predictors of ICD implantation (rather than differences between models, as is this paragraph’s focus). 3. More explanation is needed in the Methods to define ATE and ATT. Also, it is difficult as currently presented to understand the
---

	difference between results presented in Figure 3 and supplementary Figure 3. 4. Discussion, line 222 – For readability and clarity, I would suggest beginning your Discussion section with this summary of your results; move material from lines 213-221 below after you have presented your own results. 5. For all Tables and Figures, more descriptive titles, legends, and footnotes would improve readability. Minor comments:  1. In the abstract, the methods state, “We examined the association between ICD and survival using propensity score,” and later that, “Cox regression was used to model survival.” Consider some edits to clarify how the propensity score vs cox regression was used. 2. Abstract – consider adding a sentence in the Abstract Results section re: how many individuals were in the ICD and no ICD groups. 3. Pg. 4, line 68: small typo: “...superior to amiodarone for preventing a recurrent cardiac arrests.” 4. Pg. 5, line 89: “A total of 3851 with data on ICD implantation were discharged alive.” Can the authors give data separately for discharged alive vs. ICD implantation data available? E.g., something like, “A total of xxxx cases were discharged alive; of these, xxxx cases had data on ICD implantation.” 5. Pg 5, exposure: “Implantation of an ICD during hospitalization was recorded as a binary variable in the registry.” – Do the authors have information regarding the proportion of individuals with ICDs implanted after discharge (but within a reasonable time period)? 6. Pg. 6, line 115: “We removed variables with zero variance, as well as factor variables with >30 levels.” This sentence is the same as a sentence in the previous paragraph. 7. Pg. 6, “The random number generator was not fixed in order to ensure reproducibility across computations.” Do the authors mean that it WAS fixed? 8. Pg. 6, line 104: Consider editing the sentence, “A total of 919 candidate predictors were assessed.” to clarify that you are selecting predictors of ICD implantation (which is only mentioned at the end of the paragraph). 9. Pg. 8, line 162: A small typo in “...those who experienced the composite endpoint did so within 90%s” – should this be “within 90 days”? 10. Pg. 8, lines 163-164: The meaning of this sentence is unclear: “In relation to ICD implantation, of those experiencing events, 25% did so in the ICD group, compared with 52% in the group not receiving an ICD.” 11. Pg. 9, section titled “Modeling results” – this information was already given in the Methods section and could be cut here. 12. Pg. 9, lines 172-178: This paragraph seems extraneous. A detailed discussion of the differences between the 2 approaches, particularly with a focus on features that were not high on the feature importance list (e.g., ages of children at home) does not seem directly related to the study’s aims. It seems more relevant to discuss the features/predictors that were important in both models (or in the model the authors chose) for predicting ICD implantation. 13. Table 1 – Add a footnote for definition of “SMD”. 14. Table 1 – The median time for “Arrest to CPR start” is 0 and 1 minute for the non-ICD group and ICD groups respectively, arrest to ambulance arrival is 9 minutes for each group, and bystander
--	---

	CPR was performed for 54% and 79% of each group respectively. These times are difficult to understand – e.g., how does median time for CPR start equal 0 minutes for the non-ICD group when nearly half of this group did not receive CPR until ambulance arrival? 15. Figure 1 – can the authors say more in a legend or footnote to explain the vertical dashed line’s meaning in the survival probability figure? (e.g., on Pg. 8. Lines 159-161, the authors write, “The most pronounced survival trait was noted for cases who did not receive an ICD, of whom the majority of deaths occurred early after discharge (Figure 1).” – it appears that this is shown in the lines to the right of the dashed line, since survival is 96.2% for the no-ICD group there, but 92.3% for the no-ICD group to the left of the dashed line? Results will be clearer if Figure 1 is annotated with more detail. 16. Figure 2 – seems extraneous to main study aims and results. If it is retained as a main figure, it would benefit from some edits... a more specific title, a legend describing what the figure shows, a more easily interpretable feature list (e.g., “acute myocardial infarction” rather than “Comorbidity ICD code: I21_acute”). Also, the right panel is labeled “Relative Influence (ICD Exposure)” – does this show predictors for ICD implantation? 17. Figure 3 – for this figure and for the Methods and Results text for analysis of ATE and ATT, more details are needed re: how were ATE and ATT defined. Would be more accurate to present axis on logarithmic scale (i.e., distance between 1 and 0.5 should be the same as distance between 1 and 2). 18. Supp. Figure 1 and 2 – It appears that the variables shown are both positively (e.g., acute MI) and negatively (e.g., PCI performed, witnessed by ambulance) associated with ICD implantation – if so, can the authors clarify this in the figure legend? Also, what is the difference between “Bystander CPR, old variable” and “Bystander CPR”? 19. Supp. Figure 3 – Can the authors clarify whether the top of the figure shows unadjusted hazard ratios for the outcomes, and the bottom shows the adjusted hazard ratios? Also, the authors should consider plotting on a log scale such that the distance between a hazard ratio of 1 and 0.5 is the same distance as between 1 and 2. This would particularly help presentation of the HRs <1 which are difficult to evaluate as currently shown.
--	---

VERSION 1 – AUTHOR RESPONSE

Reviewer: 1

Dr. Helle Søholm, Copenhagen University Hospital Rigshospitalet

Comments to the Author:

Review ” Early ICD implantation and survival after out-of-hospital cardiac arrest”

Thank you for the opportunity to review the submitted manuscript. The study examines approx. 55.000 patients registered in the SwedHeart registry with OHCA of cardiac cause from 2010 to 2020 and the association between ICD implantation and survival. Approx. 3000 patients with available ICD-data and shockable rhythm were included the final study cohort (patients discharged alive). The result of the study is clear with a HR of 0.38 for ICD-implantation, however taking the observational and retrospective study design into account.

General comments:

Please describe in further detail how the final study population is defined. It is mentioned that “available ICD-data”, however please include data on patients with cardiac cause and shockable rhythm with missing ICD-data? My concern is that a large confounder is introduced.

First of all we are very grateful to Dr Søholm taking precious time to evaluate our manuscript. This is sincerely appreciated.

Thank you. We have now clarified this paragraph (the first paragraph in Methods).

Tables are large with many data presented. Please consider whether some data could be moved to a supplementary table.

We agree. We moved the full table to the Supplementary Material, and kept an abbreviated version in the main manuscript.

In the result section more wording is used for mentioning factors not fitted into the models than actually presenting important covariates associated with outcome after ICD-implantation. I suggest rephrasing the results section with more focus on the clinical important factors with less statistical description.

We believe it is important to clarify what and why we have removed variables from the models, and in order to clarify to the reader what actually influenced the propensity scores, we included the panel on the right hand side in Figure 2, which depicts the importance of the top 50 predictors.

The discussion is section is well-written. Please also include the DanICD study currently enrolling patients (ClinicalTrials.gov ID NCT04576130).

Very important. Thank you, it has been added. We look very much forward to this critical study.

Minor comments:

The abstract is well-written and nicely presents the scope and results of the manuscript. However, the first sentence of the conclusion “Mortality in survivors of OHCA occurs early after discharge” is not substantiated in the result section. The 90-day event rate is presented, however not how early after discharge death occurs. Please include this in the abstract.

We agree, we removed that sentence.

A conclusion section in the main manuscript seems to be missing.

Done.

Reviewer: 2

Dr. Kyndaron Reinier, Cedars-Sinai Medical Center

Comments to the Author:

Summary:

Major comments:

1. The paper's main aim, as stated in the Abstract and the last paragraph of the Introduction, was to study the impact of early ICD implantation on survival. The sections on predictors of ICD implantation are therefore somewhat unexpected and don't seem well-integrated into the paper. The authors should consider adding information in the introduction regarding why you are evaluating predictors of ICD implantation and be more explicit about that separate step in the Methods and Results.

We are sincerely grateful to Dr Reinier for taking valuable time to review our manuscript. On behalf of our team, thank you.

We had access to a total of >900 candidate predictors, and our philosophy, after creating propensity score models for many years, is that human interference in the model building tends to result in biases. Therefore we used a fully data driven approach to computing the propensity scores. An initial round of feature selection, as a means of dimensionality reduction, is warranted to reduce noise, overfitting and reduce computational burden. We then use the most important features and declare the relative importance for predicting exposure to an ICD (right-hand panel, Figure 2). We added a sentence to the relevant paragraph to clarify what we used the predictors for.

2. The "Covariate balance" section in Results could potentially be moved to a supplementary file. Its results do not seem directly related to the study's main aims. However, a paragraph could be added here that identifies the main predictors of ICD implantation (rather than differences between models, as is this paragraph's focus).

We respectfully ask the reviewer to keep this section in order to an immediate critical appraisal of the covariate balance, as all our results are dependent upon it. We hope this will be OK to keep.

3. More explanation is needed in the Methods to define ATE and ATT. Also, it is difficult as currently presented to understand the difference between results presented in Figure 3 and supplementary Figure 3.

Thank you. We have defined ATE and ATT now and provided references for more detailed discussions.

4. Discussion, line 222 – For readability and clarity, I would suggest beginning your Discussion section with this summary of your results; move material from lines 213-221 below after you have presented your own results.

Thank you. Done.

5. For all Tables and Figures, more descriptive titles, legends, and footnotes would improve readability.

Thank you. Done.

Minor comments:

1. In the abstract, the methods state, “We examined the association between ICD and survival using propensity score,” and later that, “Cox regression was used to model survival.” Consider some edits to clarify how the propensity score vs cox regression was used.

Thank you. Done.

2. Abstract – consider adding a sentence in the Abstract Results section re: how many individuals were in the ICD and no ICD groups.

Thank you. Done.

3. Pg. 4, line 68: small typo: “...superior to amiodarone for preventing a recurrent cardiac arrests.”

Thank you. Done.

4. Pg. 5, line 89: “A total of 3851 with data on ICD implantation were discharged alive.” Can the authors give data separately for discharged alive vs. ICD implantation data available? E.g., something like, “A total of xxxx cases were discharged alive; of these, xxxx cases had data on ICD implantation.”
[]

Thank you. This was also requested by Reviewer 1, and is addressed above.

5. Pg 5, exposure: “Implantation of an ICD during hospitalization was recorded as a binary variable in the registry.” – Do the authors have information regarding the proportion of individuals with ICDs implanted after discharge (but within a reasonable time period)?

Unfortunately we do not have information on that, which we have introduced as a limitation in the study.

6. Pg. 6, line 115: “We removed variables with zero variance, as well as factor variables with >30 levels.” This sentence is the same as a sentence in the previous paragraph.

7. Pg. 6, “The random number generator was not fixed in order to ensure reproducibility across computations.” Do the authors mean that it WAS fixed?

Thank you. We removed the duplicated sentence. The random number generator was not fixed since we did many re-calculations and wanted to make sure that the same results and conclusions would be observed every time.

8. Pg. 6, line 104: Consider editing the sentence, “A total of 919 candidate predictors were assessed.” to clarify that you are selecting predictors of ICD implantation (which is only mentioned at the end of the paragraph).

Thank you. Done.

9. Pg. 8, line 162: A small typo in “...those who experienced the composite endpoint did so within 90%*s*” – should this be “within 90 days”?

Thank you. Done.

10. Pg. 8, lines 163-164: The meaning of this sentence is unclear: “In relation to ICD implantation, of those experiencing events, 25% did so in the ICD group, compared with 52% in the group not receiving an ICD.” ✓

Thank you. This has been clarified now so that it states that the 25% and 52% were the proportion of events, among all events, that occurred within 90 days.

11. Pg. 9, section titled “Modeling results” – this information was already given in the Methods section and could be cut here.

Thank you. Done.

12. Pg. 9, lines 172-178: This paragraph seems extraneous. A detailed discussion of the differences between the 2 approaches, particularly with a focus on features that were not high on the feature importance list (e.g., ages of children at home) does not seem directly related to the study’s aims. It

seems more relevant to discuss the features/predictors that were important in both models (or in the model the authors chose) for predicting ICD implantation.

Thank you. Done.

13. Table 1 – Add a footnote for definition of “SMD”.

Thank you. Done.

14. Table 1 – The median time for “Arrest to CPR start” is 0 and 1 minute for the non-ICD group and ICD groups respectively, arrest to ambulance arrival is 9 minutes for each group, and bystander CPR was performed for 54% and 79% of each group respectively. These times are difficult to understand – e.g., how does median time for CPR start equal 0 minutes for the non-ICD group when nearly half of this group did not receive CPR until ambulance arrival?

Thank you. This is because we reported the medians for variables with generally scarce variance, and the CPR is provided by bystanders before EMS arrival.

15. Figure 1 – can the authors say more in a legend or footnote to explain the vertical dashed line’s meaning in the survival probability figure? (e.g., on Pg. 8. Lines 159-161, the authors write, “The most pronounced survival trait was noted for cases who did not receive an ICD, of whom the majority of deaths occurred early after discharge (Figure 1).” – it appears that this is shown in the lines to the right of the dashed line, since survival is 96.2% for the no-ICD group there, but 92.3% for the no-ICD group to the left of the dashed line? Results will be clearer if Figure 1 is annotated with more detail.

Thank you, we have expanded the figure legend to clarify this, and added annotations to the plot itself.

16. Figure 2 – seems extraneous to main study aims and results. If it is retained as a main figure, it would benefit from some edits... a more specific title, a legend describing what the figure shows, a more easily interpretable feature list (e.g., “acute myocardial infarction” rather than “Comorbidity ICD code: I21_acute”). Also, the right panel is labeled “Relative Influence (ICD Exposure)” – does this show predictors for ICD implantation?

Thank you. This figure has been revised. All predictor labels have been corrected. Panel headings have been added and annotations for interpretation were also added.

17. Figure 3 – for this figure and for the Methods and Results text for analysis of ATE and ATT, more details are needed re: how were ATE and ATT defined. Would be more accurate to present axis on logarithmic scale (i.e., distance between 1 and 0.5 should be the same as distance between 1 and 2).

Thank you. Explanations for ATE and ATT have been added, and a reference for further reading has also been added.

18. Supp. Figure 1 and 2 – It appears that the variables shown are both positively (e.g., acute MI) and negatively (e.g., PCI performed, witnessed by ambulance) associated with ICD implantation – if so, can the authors clarify this in the figure legend? Also, what is the difference between “Bystander CPR, old variable” and “Bystander CPR”?

This has been clarified in the figure, thank you.

19. Supp. Figure 3 – Can the authors clarify whether the top of the figure shows unadjusted hazard ratios for the outcomes, and the bottom shows the adjusted hazard ratios? Also, the authors should consider plotting on a log scale such that the distance between a hazard ratio of 1 and 0.5 is the same distance as between 1 and 2. This would particularly help presentation of the HRs <1 which are difficult to evaluate as currently shown.

This has been clarified in the figure, thank you.

VERSION 2 – REVIEW

REVIEWER	Reinier, Kyndaron Cedars-Sinai Medical Center, Smidt Heart Institute
REVIEW RETURNED	12-Nov-2023

GENERAL COMMENTS	Summary: Thank you for the opportunity to review this revised manuscript. The authors have responded adequately to most of the comments in the first review. However, in careful reading of the authors’ responses and in a re-reading of the revised manuscript, I realize now that in my earlier review I didn’t fully comprehend the authors’ description of the methods regarding propensity score weighting. Therefore, I have a few more questions regarding the propensity score analysis and how it affects the interpretation of the study’s results. I believe a few additional edits could make this manuscript and its clinical implications more understandable to readers. Major comments: 1. The authors state in the Abstract that “Treatment associations were estimated using propensity scores.” That is, they calculated propensity scores for receiving an ICD using a variety of methods, chose the method that achieved the best balance, and then used those propensity scores to generate weights (inverse probability of treatment – i.e., receiving an ICD) in the Cox regression models. Is this correct?
---

	2. If correct, can the authors provide a sentence or two in the Methods to explain why this approach was chosen for their research question? 3. Assuming the above is a correct understanding of the Methods, is it correct that the results of this analysis should be interpreted as the effect of early ICD implantation on time to event if implantation of the ICD was randomized? If so, can the authors include a statement to this effect in the Methods and also the Discussion? 4. If the above understanding is not correct, can the authors clarify how the results should be interpreted given the use of the propensity score approach? 5. Could the authors also include results of unadjusted hazard ratios for the effect of early ICD implantation on time to event from Cox proportional hazards models? 6. Can the authors clarify whether the Kaplan-Meier and Density plots in Figure 2 are from unweighted data? 7. For the subgroup analyses in Figure 3, can the authors comment on how the inclusion of weights in the Cox model impacts estimates of the ATE and ATT for subgroups that were used to generate the weights? (e.g., heart failure is a subgroup in Figure 3, but history of heart failure also had a relatively high importance for predicting ICD placement in the propensity score estimates – how does the fact that history of heart failure is used in weighting affect the estimate of the association of heart failure with the outcome?). 8. Finally, case inclusion criteria remain somewhat unclear in the Methods section. The authors state in the Abstract methods that “All cases discharged alive during 2010 to 2020 with a cardiac etiology or initial shockable rhythm were included.” In the manuscript’s Methods section, they write, “This included ... 54568 ... first events... A total of 10836 cases had information on ICD, of whom 3851 were discharged alive. Among these 3851 patients we included those with cardiac etiology or initial shockable rhythm, resulting in a final study population of 3005 patients.” It would be helpful, if the authors can do so, to indicate the number of survivors with cardiac etiology or initial shockable rhythm out of the 54568 total, and then indicate the number of this subset for whom ICD information was available. If ICD information is available for most survivors, potential bias is not a large concern. If ICD information is missing for a large number of survivors, a brief comparison in the Results section of survivors with and without ICD data available - e.g., age, sex - would indicate whether the subgroup analyzed was representative of the whole target population. Minor comment: 1. This sentence (pg. 13, line 226) needs some grammatical corrections: “Current clinical guidelines suggests that patients developing ventricular arrhythmias due to a cardiac event, in which the underlying condition resolves does not have an increased risk for future SCA, thus not eligible for ICD therapy.”
--	---

VERSION 2 – AUTHOR RESPONSE

1. The authors state in the Abstract that “Treatment associations were estimated using propensity scores.” That is, they calculated propensity scores for receiving an ICD using a variety of methods, chose the method that achieved the best balance, and then used those propensity scores to generate weights (inverse probability of treatment – i.e., receiving an ICD) in the Cox regression models. Is this correct?

That is perfectly correct.

2. If correct, can the authors provide a sentence or two in the Methods to explain why this approach was chosen for their research question?

We have added the following sentence in the methods section, line 158: *“The purpose of our approach to estimate the final propensity score was to consider all available variables, identify the most important ones and then consider several different prediction models to finally obtain the maximum covariate balance. Ultimately, this allows us to estimate the effect of having an ICD at discharge, on the risk of the outcomes (composite of death, recurrent OHCA or IHCA), as can be estimated using observational data.”*

3. Assuming the above is a correct understanding of the Methods, is it correct that the results of this analysis should be interpreted as the effect of early ICD implantation on time to event if implantation of the ICD was randomized? If so, can the authors include a statement to this effect in the Methods and also the Discussion?

Please see our answer to the previous question. Additionally we added this to the discussion section, line 311: *“Even with these considerations, the propensity score, especially when generated through machine learning techniques, stands out as a leading method for imitating a randomized controlled trial using observational data[18,19]. This approach is especially pertinent when utilizing real-world data that encompasses the whole population targeted for inferences, as in this national registry.”*

4. If the above understanding is not correct, can the authors clarify how the results should be interpreted given the use of the propensity score approach?

Please see above.

5. Could the authors also include results of unadjusted hazard ratios for the effect of early ICD implantation on time to event from Cox proportional hazards models?

Unadjusted hazard ratios were slightly larger than the adjusted ones, but with greater variance across subgroups and this is explained by the variance in the unbalanced covariates; the direction on the unbalance varies in different subgroups. However, we only aim to provide balanced estimates which gives a very consistent view of the effect of ICD implantation.

6. Can the authors clarify whether the Kaplan-Meier and Density plots in Figure 2 are from unweighted data?

Yes, this is correct.

7. For the subgroup analyses in Figure 3, can the authors comment on how the inclusion of weights in the Cox model impacts estimates of the ATE and ATT for subgroups that were used to generate the weights? (e.g., heart failure is a subgroup in Figure 3, but history of heart failure also had a relatively high importance for predicting ICD placement in the propensity score estimates – how does the fact that history of heart failure is used in weighting affect the estimate of the association of heart failure with the outcome?).

The subgroups only represented COX-regressions performed in a subset of the original population. The subgroups did not affect the weights and intend only to present the hazard ratios in relevant subgroups.

8. Finally, case inclusion criteria remain somewhat unclear in the Methods section. The authors state in the Abstract methods that “All cases discharged alive during 2010 to 2020 with a cardiac etiology or initial shockable rhythm were included.” In the manuscript’s Methods section, they write, “This included ... 54568 ... first events... A total of 10836 cases had information on ICD, of whom 3851 were discharged alive. Among these 3851 patients we included those with cardiac etiology or initial shockable rhythm, resulting in a final study population of 3005 patients.” It would be helpful, if the authors can do so, to indicate the number of survivors with cardiac etiology or initial shockable rhythm out of the 54568 total, and then indicate the number of this subset for whom ICD information was available. If ICD information is available for most survivors, potential bias is not a large concern. If ICD information is missing for a large number of survivors, a brief comparison in the Results section of survivors with and without ICD data available - e.g., age, sex - would indicate whether the subgroup analyzed was representative of the whole target population.

For the population you are referring to, the following figure show the missingness for ICD implantation.

We note that the rate of missingness for ICD implantation was around 15% among survivors with cardiac etiology. Missingness may also affect the robustness of our estimates, although we argue that the pronounced and consistent reduction of the outcomes cannot be explained by missingness which is more likely to be due to randomness.

Minor comment:

1. This sentence (pg. 13, line 226) needs some grammatical corrections: “Current clinical guidelines suggests that patients developing ventricular arrythmias due to a cardiac event, in which the underlying condition resolves does not have an increased risk for future SCA, thus not eligible for ICD therapy.”

We have changed this, refer to line 263 in the manuscript.

VERSION 3 – REVIEW

REVIEWER	Reinier, Kyndaron Cedars-Sinai Medical Center, Smidt Heart Institute
REVIEW RETURNED	08-Jan-2024
GENERAL COMMENTS	The authors have responded to all previous comments adequately. There is one minor remaining issue: In your response to reviewers, comment #8, you wrote: “We note that the rate of missingness for ICD implantation was around 15% among survivors with cardiac etiology.” In your edited manuscript, the

	phrase “rate of missingness for ICD implantation” became “rate for ICD implantation,” (line 315, pasted below) changing its meaning. Please check and correct. “In regard to missing data, we found that the rate for ICD implantation was around 15% among survivors with cardiac etiology.”
--	---